# p32-Dependent p38 MAPK Activation by Arginase II Downregulation Contributes to Endothelial Nitric Oxide Synthase Activation in HUVECs

**DOI:** 10.3390/cells9020392

**Published:** 2020-02-08

**Authors:** Bon-Hyeock Koo, Moo-Ho Won, Young-Myeong Kim, Sungwoo Ryoo

**Affiliations:** 1Department of Biological Sciences, Kangwon National University, Chuncheon, Gangwon 24341, Korea; arsk123@naver.com; 2Department of Neurobiology, School of Medicine, Kangwon National University, Chuncheon, Gangwon 24341, Korea; mhwon@kangwon.ac.kr; 3Department of Molecular and Cellular Biochemistry, School of Medicine, Kangwon National University, Chuncheon, Gangwon 24341, Korea; ymkim@kangwon.ac.kr

**Keywords:** arginase II, calcium, endothelial nitric oxide synthase, p38 MAPK, p32

## Abstract

Arginase II reciprocally regulates endothelial nitric oxide synthase (eNOS) through a p32-dependent Ca^2+^ control. We investigated the signaling pathway of arginase II-dependent eNOS phosphorylation. Western blot analysis was applied for examining protein activation and [Ca^2+^]c was analyzed by microscopic and FACS analyses. Nitric oxide (NO) and reactive oxygen species (ROS) productions were measured using specific fluorescent dyes under microscopy. NO signaling pathway was tested by measuring vascular tension. Following arginase II downregulation by chemical inhibition or gene knockout (KO, ArgII^−/−^), increased eNOS phosphorylation at Ser1177 and decreased phosphorylation at Thr495 was depend on p38 MAPK activation, which induced by CaMKII activation through p32-dependent increase in [Ca^2+^]c. The protein amount of p32 negatively regulated p38 MAPK activation. p38 MAPK contributed to Akt-induced eNOS phosphorylation at Ser1177 that resulted in accelerated NO production and reduced reactive oxygen species production in aortic endothelia. In vascular tension assay, p38 MAPK inhibitor decreased acetylcholine-induced vasorelaxation responses and increased phenylephrine-dependent vasoconstrictive responses. In ApoE^−/−^ mice fed a high cholesterol diet, arginase II inhibition restored p32/CaMKII/p38 MAPK/Akt/eNOS signaling cascade that was attenuated by p38 MAPK inhibition. Here, we demonstrated a novel signaling pathway contributing to understanding of the relationship between arginase II, endothelial dysfunction, and atherogenesis.

## 1. Introduction

Nitric oxide (NO) is an endothelium-dependent vasorelaxant that plays a major role in cardiovascular physiology. NO is a potent vasodilator and essential in the regulation of vascular tone and blood pressure, and additionally, NO regulates homeostasis by preventing platelet and leukocyte adhesion, and proliferation and migration of smooth muscle cells. Endothelial dysfunction resulting from a decrease in NO bioavailability, impaired NO signaling, and an increase in reactive oxygen species (ROS), has been identified as an early sign of multiple diseases and vascular pathology [1].

p38 Mitogen-activated protein kinases (MAPKs) are members of the mitogen-activated serine/threonine kinase family and can be activated by pathogenic stimuli, extracellular stress, and intracellular stress. We previously demonstrated that p38 MAPK activation plays a key role in interleukin-8 production in native low-density lipoprotein (nLDL)-stimulated human aortic smooth muscle cells (hAoSMCs) and AoSMCs of ApoE^−/−^ mice fed a high cholesterol diet (HCD) [2]. An increased mitochondrial calcium concentration ([Ca^2+^]m) has been associated with p38 MAPK activation in AoSMCs from humans and mice, and nLDL stimulation has led to a p32-dependent increase in [Ca^2+^]m and a decrease in cytosolic calcium concentration ([Ca^2+^]c). Activation of p38 MAPK and changes in intracellular [Ca^2+^] were prevented by inhibiting arginase II with the arginase inhibitor 2(S)-amino-6-boronohexanoic acid (ABH) or siRNA [3]. Although it has been demonstrated that inhibition of arginase II activates endothelial nitric oxide synthase (eNOS), the role of p38 MAPK in eNOS activation is unclear.

Arginase II, the extrahepatic isoform of arginase, is primarily localized to the mitochondria and is the principal form in human and mouse aortic endothelial cells (ECs). Arginase II catalyzes the conversion of L-arginine (L-arg) to urea and L-ornithine, which is the precursor for the synthesis of the polyamines putrescine, spermidine, and spermine that are associated with cell differentiation and proliferation [4,5]. Arginase II expression and activity is upregulated in many disorders, including aging [6], ischemia-reperfusion [7,8], hypertension [9,10], balloon injury [11], and atherosclerosis [11], and can be induced by hypoxia, lipopolysaccharides, and tumor necrosis factor-α [12,13,14]. Arginases have been repeatedly shown to reciprocally regulate NO production [6,11,15]. Arginase-dependent decline in NO production involves translocation of arginase II from the mitochondria to the cytosol, which leads to L-arg depletion and eNOS uncoupling [16]. Recently, we suggested a novel mechanism in which arginase II regulates [Ca^2+^]c via p32, which is involved in eNOS activation [15]. However, the signaling cascade leading to eNOS phosphorylation by arginase II downregulation has not yet been elucidated.

The important physiological effects of Akt on apoptosis, cell attachment, and cell proliferation can be attributed to an increase in NO production through Akt-dependent eNOS activation [17,18,19]. Two distinct mechanisms, Ca^2+^-dependent [20] and Ca^2+^-independent (such as PI(3)K-dependent) [21], can lead Akt activation. The phosphorylation of Akt at Thr 308 and Ser 473 can exert fully activation and augment NO production via phosphorylation of eNOS at Ser 1177 [22,23].

Therefore, we first examined the activation of p38 MAPK in arginase II-downregulated HUVECs and ArgII-null mice and the involvement of p38 MAPK in eNOS activation. Because arginase II downregulation induced a p32-dependent increase in [Ca^2+^]c, p32- and Ca^2+^-dependent p38 MAPK activation were investigated. Additionally, we tested whether Akt activation was associated with p38 MAPK-dependent eNOS activation. Finally, the signaling cascade, ArgII downregulation CaMKII/p38 MAPK/Akt/eNOS phosphorylation, was confirmed in atherogenic model ApoE^−/−^ mice fed an HCD.

## 2. Materials and Methods

### 2.1. Materials

The following chemicals: 2(*S*)-amino-6-boronohexanoic acid (ABH), *N*^G^-nitro-l-arginine methyl ester (L-NAME), manganese (III) tetrakis (4-benzoic acid) porphyrin chloride (MnTBAP), KN-93, Akti-1/2, and SB202190 were purchased from Calbiochem (Darmstadt, Germany). All other chemicals were obtained from Sigma-Aldrich (St. Louis, MO, USA) unless otherwise stated. Antisera against eNOS (Catalog No. 610296), phospho-eNOS (Ser1177 (Catalog No. 612392) and Thr495 (Catalog No. 612707)), phospho-CaMKII (Catalog No. 12716), phospho-p38 MAPK (Catalog No. 9211), p38 MAPK (Catalog No. 9212), phospho-Akt (Ser473 (Catalog No. 9271) and Thr308 (Catalog No. 9275)), and pan-actin (Catalog No. 4968) were obtained from BD Biosciences (San Jose, CA), and p32 antiserum (Catalog No. ab24733) was obtained from Abcam (Cambridge, UK). Antiserum against voltage-dependent anion channel (VDAC), siRNA targeting arginase II (siArgII, sc-29729), and scrambled siRNA (scmRNA, sc-37007) were purchased from Santa Cruz Biotechnology (Santa Cruz Biotech, CA, USA).

### 2.2. Cell Culture and Animals

HUVECs were purchased from Cascade Biologics (Portland, OR) and were maintained according to the supplier’s instructions. Ten-week-old male C57BL/6J wild-type (WT) and male ApoE^−/−^ mice (Daehan Biolink Co. Chungbuk, Korea) were fed a normal diet (ND) or an HCD (D12108C, Research Diets, New Brunswick, NJ) for eight weeks. Arginase II knockout (KO, ArgII^−/−^) mice breeders with C57BL/6 background were a generous gift from Professor Jate P.F. Chin-Dusting for establishment of a colony at the Kanwon National University Animal Services Facility and 10-week-old male mice were used for all experiments. Arginase II knockout (KO, ArgII^−/−^) mice and eNOS KO mice with a C57BL/6 background were maintained at the Kangwon National University Animal Services Facility. All mice used in the experiment were bred in a permitted breeding room to maintain the same condition. Mice were housed at 23 °C under a 12 h light/12 h dark cycle. All animals had access to water and food (Nara Biotech. Korea) ad libitum. This study was approved in accordance with the Guide for the Care and Use of Laboratory Animals (Institutional Review Board, Kangwon National University). This investigation conformed to the principles outlined in the Declaration of Helsinki.

### 2.3. Western Blot Analysis

Cell lysates and aortic lysates were subjected to SDS-PAGE followed by Western blot, as previously described [11]. Band intensities were analyzed using NIH ImageJ software.

### 2.4. [Ca^2+^]c Measurement using Confocal Microscopy and Flow Cytometry

[Ca^2+^]c was monitored using Fluo-4 AM (100 nmol/L, 1 h, Thermo Fisher Scientific, Waltham, MA) with excitation at 494 nm, and emission at 506 nm. Intensity values were normalized to the initial fluorescence values after subtraction of background using Metamorph (Molecular Probes, San Jose, CA, USA). [Ca^2+^]c was also determined using flow cytometry (FACSCalibur, BD Biosciences, San Jose, CA) and fluorescence intensity for each sample was determined using CellQuest software (BD Biosciences). Ca^2+^ levels were evaluated by determining the fold change in fluorescence intensity of treated versus control cells.

### 2.5. Mitochondrial Fractionation

Cells were homogenized twice in subcellular fractionation buffer containing 250 mmol/L sucrose, 20 mmol/L HEPES pH 7.4, 10 mmol/L KCl, 1.5 mmol/L MgCl_2_, 1 mmol/L EDTA, 1 mmol/L EGTA, and Roche protease inhibitors (Sigma-Aldrich) for 3 min and centrifuged at 1000× *g* for 10 min to remove cell debris and unlysed cells. Supernatants were centrifuged at 21,000× *g* for 45 min at 4 °C. Cytosolic (supernatant) and mitochondrial (precipitate) fractions containing 20 μg of total proteins were used for Western blot analysis of p32 protein expression.

### 2.6. p32 Plasmid and siRNA Transfection

For siRNA transfection, HUVECs were incubated in starvation medium (DMEM plus 5% FBS and antibiotics) containing an siRNA targeting p32 (sip32, 100 nmol/L, 5′-TGT CTC CGT CGG TGT GCA GC-Cy5- 3′), scrambled siRNA (scmRNA, 100 nmol/L, 5′-GCT GCA CAC CGA CGG AGA CA-Cy5-3′), or no oligonucleotide for 24 h without a reagent. HUVECs were cultured in 6-well plates and were transfected with 1 μg of the pCMV6-XL5-p32 plasmid (OriGene, SC107905, Rockville, MD, USA) or the empty plasmid of pCMV6-XL5 using Lipofectamine 3000 (Thermo Fisher Scientific). After 6 h of incubation, the cells were cultured for another 24 h in fresh growth medium.

### 2.7. Measurement of NO and ROS

Aortic rings from 10-week-old male C57BL/6 WT mice were labeled for superoxide detection with 1 μmol/L dihydroethidine (DHE) for 5 min with 30 S intervals or were labeled for NO with 5 μmol/L 4-amino-5-methylamino-2’,7´-difluorofluorescein diacetate (DAF-FM DA) for 5 min with 30 S intervals. Images were acquired using an Olympus BX51 epifluorescence microscope. Fluorescence intensity was measured, as previously described [11], using Metamorph software.

### 2.8. Vascular Tension Assay

Heparin was administered 1 h before mice were sacrificed. Mice were anesthetized using isoflurane, and the thoracic aorta from the aortic root to the bifurcation of the iliac arteries was rapidly isolated and cut into 1.5 mm rings. The aortic rings were placed in ice-cold oxygenated Krebs-Ringer bicarbonate buffer (118.3 mmol/L NaCl, 4.7 mmol/L KCl, 1.2 mmol/L MgSO_4_, 1.6 mmol/L CaCl_2_, 25 mmol/L NaHCO_3_, 11.1 mmol/L glucose, pH 7.4) and suspended between two wire stirrups (150 mm) in a myograph (Multi Myograph System, DMT-620, Hinnerup, Denmark) containing 10 mL of Krebs-Ringer (95% O_2_ and 5% CO_2_, pH 7.4, 37 °C). One stirrup was connected to a three-dimensional micromanipulator, and the other to a force transducer. The aortic rings were passively stretched at 10 min intervals in increments of 100 mg to reach the optimal tone of 600 mg. After stretching to 600 mg, the contractile response to 60 mmol/L KCl was determined. The response to a maximal dose of KCl was used to normalize the responses to agonist across vessel rings. Dose responses to the vasoconstrictor phenylephrine (PE, 10^–9^ to 10^–5^ mol/L) were assessed, and responses to the vasodilators acetylcholine (Ach, 10^–9^–10^–5^ mol/L) and sodium nitroprusside (SNP, 10^–10^ to 10^–6^ mol/L) were assessed after preconstriction with PE (10^–5^ mol/L). To further confirm the NO-dependent vasorelaxation activity, aortic rings were treated with 1H-[1,2,4]oxadiazolo[4,3-a]quinoxalin-1-one (ODQ, 10^–5^ mol/L), a soluble guanylyl cyclase inhibitor.

### 2.9. Statistical Methods

All experiments were performed with three to four biological replicates, and the specific number of replicates is reported for each experiment. Student’s *t*-tests and two-way ANOVA were performed as appropriate (GraphPad Prism, GraphPad Software, San Diego, CA). Values were plotted as arithmetic mean ± standard error of the mean (SE). Significance levels were determined and a *p* < 0.05 was considered statistically significant.

## 3. Results

### 3.1. p38 MAPK Activation Plays a Key Role in eNOS Phosphorylation at Ser1177 Following Arginase II Downregulation

We first tested p38 MAPK activation and eNOS Ser1177 phosphorylation in intact (+ECs) and de-endothelialized (−ECs) aortic vessels from WT and ArgII^−/−^ mice. The p38 MAPK activation and eNOS Ser1177 phosphorylation were enhanced, but eNOS Thr495 phosphorylation was reduced in the endothelia of ArgII^−/−^ mice as compared with those of WT mice (Figure 1A,B). Treatment with ABH for 30 min activated p38 MAPK (Figure 1C) and phosphorylated eNOS at Ser1177, but decreased phosphorylation of eNOS at Thr495 (Figure 1D). To examine the role of p38 MAPK in eNOS Ser1177 phosphorylation mediated by arginase II downregulation, HUVECs incubated with ABH or siArgII and, subsequently, treated with the p38 MAPK inhibitor SB202190. SB202190 prevented eNOS Ser1177 phosphorylation and augmented phosphorylation of eNOS at Thr495 (Figure 1E,F). The p38 MAPK inhibition with SB202190 blunted eNOS Ser1177 phosphorylation in aortas of both WT and ArgII^−/−^ and enhanced eNOS Thr495 phosphorylation (Figure 1G).

### 3.2. Arginase II Downregulation Induces CaMKII-Dependent p38 MAPK Activation through Increased [Ca^2+^]c

Arginase II downregulation has been shown to increases [Ca^2+^]c [15], and we confirmed this effect by microscopy and FACS analysis upon downregulating arginase II using ABH or siArgII (Figure 2A,B). Because increased [Ca^2+^]c is associated with CaMKII activation, we tested the effect of the CaMKII inhibitor, KN-93. We found that KN-93 blocked ABH-dependent p38 MAPK activation, but that inhibition of p38 MAPK did not affect ABH-dependent CaMKII phosphorylation (Figure 2C). Thus, p38 MAPK activation induced by arginase II downregulation depends on the CaMKII activation via increased [Ca^2+^]c.

### 3.3. p32 Expression Regulates p38 MAPK Activation

We previously reported that arginase II regulates mitochondrial and cytosolic [Ca^2+^] via mitochondrial p32. High p32 expression reduced [Ca^2+^]c and treatment with sip32 increased [Ca^2+^]c [15]. Downregulation of p32 by sip32 activated p38 MAPK (Figure 3A,B), and p32 overexpression by plasmid transfection reduced phosphorylation of p38 MAPK (Figure 3C,D). Arginase inhibition with ABH had no effect on sip32-dependent p38 MAPK phosphorylation (Figure 3B), however, recovered p38 MAPK phosphorylation attenuated by p32 overexpression (Figure 3D). Thus, mitochondrial p32 negatively regulates p38 MAPK activation.

### 3.4. Activated p38 MAPK Induces Phosphorylation of eNOS at Ser1177 through Akt Activation

We examined the role of Akt activation in eNOS phosphorylation caused by arginase II downregulation. Treatment with ABH or siArgII increased Akt phosphorylation at Ser473 and Thr308 (Figure 4A,B). The p38 MAPK inhibitor SB202190 blocked Akt phosphorylation caused by arginase II downregulation (Figure 4C). Akt was inhibited with Akti-1/2, and this inhibition prevented the phosphorylation of eNOS at Ser1177 by arginase II downregulation (Figure 4D). Therefore, p38 MAPK/Akt activation is required for eNOS phosphorylation at Ser1177. Furthermore, we investigated the signaling cascade eNOS phosphorylation in ABH-treated HUVECs. As shown in Appendix A, arginase inhibition induced activation of CaMKII/AMPK/p38 MAPK/Akt signaling pathway. Finally, this pathway was associated with increased eNOS phosphrylation at Ser1177 and Ser617 and reduced eNOS phosphrylation at Thr495 and Ser114 (Appendix A). We further tested that an increased NO level did not have an effect on this cascade using eNOS-null mice (Appendix A).

### 3.5. p38 MAPK Regulates NO Production and ROS Generation and Inhibition of p38 MAPK Impairs Endothelium-Dependent Vasorelaxation in Arginase II-Downregulated Aortic Vessels

Because p38 MAPK activation by arginase II downregulation increased eNOS Ser1177 phosphorylation, we tested whether p38 MAPK affects NO production and ROS generation in aortic endothelia of WT and ArgII^−/−^ mice. Arginase downregulation, ABH treatment and ArgII^−/−^ mice, enhances NO production that was suppressed with the p38 MAPK inhibitor SB203190 (Figure 5A) in endothelia. However, the reduced ROS generation by arginase II downregulation was reversed treated with SB203190 (Figure 5B). To confirm the effect of p38 MAPK on NO signaling in the aortas, we performed a vascular tension assay. The accumulated dose responses to the endothelium-dependent vasorelaxant Ach were higher in the aortas of ArgII^−/−^ mice than in those of WT mice (WT vs. ArgII^−/−^, Emax, 74.6% ± 1.05% vs. 98.4% ± 1.09%, *p* < 0.05) that was significantly attenuated by the p38 MAPK inhibitor (Figure 5C, ArgII^−/−^ vs. ArgII^−/−^, +SB202190, Emax, 98.4% ± 1.09% vs. 76.5% ±1.59%, *p* < 0.05), but SNP responses did not differ among the groups (Figure 5D, Emax, WT = 100.2% ± 0.37%, WT + SB202190 = 98.7% ± 0.83%, ArgII^−/−^ = 97.6% ± 0.94%, ArgII^−/−^ + SB202190 = 100.4% ± 1.38%, not significant). Despite the responses to PE were retarded in untreated aortas from ArgII^−/−^ mice (Emax, WT vs. ArgII^−/−^ = 135.9% ± 3.13% vs. 48.6% ± 1.33%, *p* < 0.05), dose responses to the vasoconstrictor PE were enhanced in SB202190-treated aortas of ArgII^−/−^ mice, (Figure 5E, ArgII^−/−^ vs. ArgII^−/−^ + SB202190, Emax, 48.6% ± 1.33% vs. 147.2% ± 7.77%, *p* < 0.05). Incubation of aortas with the sGC inhibitor ODQ induced similar levels of vessel constriction in all of the groups (Figure 5F, WT, WT+SB202190, ArgII^−/−^, ArgII^−/−^+SB202190, Emax, 62.7% ± 3.8%, 61.6% ± 5.4%, 60.9% ± 1.9%, 63.5% ± 7.0%). Consistent with these results, vasorelaxation responses to Ach in ABH-treated aortas of WT mice were attenuated by treatment with p38 MAPK inhibitor (Figure 5G, WT, WT+ABH, WT+ABH+SB202190, Emax, 82.8% ± 4.1%, 98.8% ± 2.3%, 76.5% ± 2.5%), but, SNP responses did not change (Figure 5H, WT, WT+ABH, WT+ABH+SB202190, Emax, 101.4% ± 1.5%, 101.7% ± 1.8%, 94.6% ± 2.7%). PE-induced constrictive responses in ABH-treated aortas were enhanced by treatment with the p38 MAPK inhibitor (Figure 5I, WT, WT+ABH, WT+ABH+SB202190, Emax, 103.4% ± 2.8%, 82.8% ± 3.8%, 110.3% ± 2.5%), however, the ODQ-dependent constrictive responses did not differ among the groups (Figure 5J, WT, WT+ABH, WT+ABH+SB202190, Emax, 62.7% ± 3.8%, 61.6% ± 5.4%, 65.3% ± 9.6%). These results indicate that p38 MAPK activation in both untreated-normal vessel and arginase II downregulated-vessel are involved in enhanced NO signal transduction.

### 3.6. p38 MAPK Inhibition Blocks Arginase II-Dependent eNOS Phosphorylation in ApoE^−/−^ Mice fed an HCD

We next investigated the role of p38 MAPK in eNOS activation by arginase II inhibition in atherogenic ApoE^−/−^ mice fed an HCD. In vessels of WT and ApoE^−/−^ mice fed a ND, p38 MAPK inhibition significantly attenuated Akt/eNOS activation without affecting CaMKII phosphorylation. In the aortas of ApoE^−/−^ mice fed an HCD, the CaMKII/p38 MAPK/Akt/eNOS Ser1177 signaling cascade was significantly attenuated, and this attenuation was reversed by incubation with ABH. The beneficial effects of ABH were lost upon inhibition of p38 MAPK (Figure 6). However, in ABH-treated ApoE^−/−^ mice fed an HCD, CaMKII phosphorylation was slightly reduced upon treatment of p38 MAPK inhibitor. To address this, we examined CaMKII phosphorylation in de-endothelialized aortic vessels and found that treatment with SB202190 markedly reduced CaMKII phosphorylation (Appendix A). Attenuation of CaMKII phosphorylation by p38 MAPK inhibition seems to depend on the presence of vascular smooth muscle cells, which indicates that the signaling cascade between p38 MAPK and CaMKII differs in ECs and vascular smooth muscle cells. Interestingly, p32 protein levels increased in aortas of ApoE^−/−^ mice fed an HCD.

### 3.7. Restored Vessels Reactivity by Arginase II Inhibition is Lost with p38 MAPK Inhibition in ApoE^−/−^ Mice Fed an HCD

We measured endothelial NO production using DAF fluorescence. The p38 MAPK inhibition reduced NO production in WT and ApoE^−/−^ mice fed a normal diet. Attenuated NO production in ApoE^−/−^ mice fed an HCD was restored with treatment with ABH that was blocked by treatment with SB202190 (Figure 7A). Ach-induced relaxation responses in the aortas of ApoE^−/−^ mice fed an HCD were reduced as compared with WT mice, and SB202190 further attenuated vessels relaxation in all groups (Figure 7B), but responses to treatment with SNP did not differ among the groups (Figure 7C). Vasoconstrictive responses to PE were enhanced in aortas of ApoE^−/−^ mice fed an HCD as compared with those of WT mice, and SB202190 treatment augmented this response (Figure 7D). The impaired vasorelaxation responses to Ach in aortas of ApoE^−/−^ mice fed an HCD were improved when the aortas were treated with ABH, but the improvement was lost upon treatment with SB202190 (Figure 7E). In aortas of ApoE^−/−^ mice fed an HCD, the constrictive responses to PE were improved upon arginase inhibition, but this improvement was lost with p38 MAPK inhibition (Figure 7F). Vasorelaxation responses to SNP did not differ between the groups (Figure 7G).

## 4. Discussion

In this study, we showed that downregulation of arginase II with ABH or siRNA increased phosphorylation of eNOS at Ser1177 and decreased phosphorylation of eNOS at Thr495 through activation of p38 MAPK, which was activated by Ca^2+^-dependent CaMKII. In addition, arginase II was found to regulate [Ca^2+^]c in a p32-dependent manner, thus, p32 expression was associated reciprocally with p38 MAPK activation. Activated p38 MAPK induced Akt-dependent phosphorylation of eNOS at Ser1177. The p38 MAPK inhibition blocked enhanced NO production by arginase II downregulation in aortic endothelia, whereas ROS generation was augmented by p38 MAPK inhibition. In the vascular tension assay, p38 MAPK inhibition alleviated the arginase II-dependent enhancement of the Ach-dependent vasorelaxation responses and exacerbated the PE-dependent vasoconstrictive responses. In atherogenic mice fed an HCD, arginase II inhibition by ABH restored the CaMKII/p38 MAPK/Akt/eNOS signaling cascade that impaired with p38 MAPK inhibition without affecting CaMKII activation and the p32 protein level was increased. The Ach-induced relaxing activity and PE-induced constrictive responses in aortas of ApoE^−/−^ fed an HCD were recovered with ABH incubation that was impaired by p38 MAPK inhibition.

Studies have identified p32, also called HABP1 (hyaluronan-binding protein 1) or C1qbp (globular head domains complement 1q) as a subunit of the pre-mRNA splicing factor SF2 in the nucleus [24] and as a receptor that interacts with complement component C1q on the cell surface [25]. Recently, we showed that p32 is predominantly targeted to mitochondria in ECs [16]. Along with the roles of p32 in the induction of cell death [26,27] and in the metabolic shift between oxidative phosphorylation and aerobic glycolysis [28], we reported that p32 played an important protein in the regulation of Ca^2+^ levels between mitochondria and cytosol [15]. Interestingly, p32-dependent Ca^2+^ regulation was shown to be controlled by free spermine level, which is affected by arginase II activity, and p32 protein stability was shown to be regulated by the arginase II protein level. Although we showed that p32 downregulation increased [Ca^2+^]c, which is involved in the eNOS phosphorylation, the signaling cascade of p32-dependent eNOS activation remained unclear. Here, we presented that overexpression of p32 using plasmid transfection technique decreased p38 MAPK phosphorylation and that knockdown of p32 by sip32 activated p38 MPAK (Figure 3). Activated p38 MAPK induced eNOS phosphorylation in an Akt-dependent manner (Figure 4). This signaling cascade was also observed in the aortas of atherogenic ApoE^−/−^ mice (Figure 7). In the ApoE^−/−^ mice fed an HCD, the p32 protein level was increased and p38 MAPK activation was blunted, resulting in eNOS inactivation. However, arginase II inhibition by ABH restored the signaling pathway of [Ca^2+^]c/CaMKII/p38 MAPK/Akt/eNOS activation, whereas p38 MAPK inhibiton prevented eNOS activation. To clearly demonstrate the signaling cascade, chemical inhibitors were used to identify the kinases involved between arginase II inhibition and eNOS phosphorylation and CaMKII was found to activate AMPK, which is involved in p38 MAPK activation (Appendix A). Because eNOS activity can be regulated by multi-site phosphorylation [29], we examined the changes of eNOS phosphorylation at different sites (Appendix A). Arginase II inhibition increased phosphorylation of eNOS at Ser1177 and Ser617 and decreased phosphorylation of eNOS at Thr495 and Ser114 in a time-dependent manner. These results were further confirmed by measuring NO levels in aortic endothelia and by the vascular tension assay (Figure 7). Therefore, our data indicate that p38 MAPK activation is essential for the improvement of endothelial dysfunction caused by arginase II upregulation.

Ca^2+^ is a principal modulator of eNOS activity. Increased [Ca^2+^]c activates calmodulin (Ca^2+^/CaM). Activated Ca^2+^/CaM binds to the canonical CaM-binding domain in eNOS and facilitates electron flow from the reductase domain to oxygenase domain by displacing the adjacent autoinhibitory loop of eNOS, thereby, preventing eNOS Thr495 phosphorylation and leading to efficient NO synthesis. Ca^2+^/CaM activates Ca^2+^/CaMKII, which leads to phosphorylation of eNOS Ser1177 in the reductase domain and enhances electron flux and NO production [30,31]. However, the identities of the protein kinases and phosphatases that stimulate eNOS activity by Ser1177 phosphorylation and Thr495 dephosphorylation remains controversial and could be coordinated independently [32]. The kinases AMPK, PKA, Akt, and PKC and the phosphatases, PP1 and PP2A have been suggested based on their multiple stimuli [21,33,34,35]. This study shows that elevated intracellular Ca^2+^ by arginase II inhibition regulates eNOS phosphorylation in a CaM-dependent manner. Activated CaM plays two important roles in eNOS phosphorylation as follows: (1) Activated Ca^2+^/CaM bound to Ca^2+^/CaM-binding domain of eNOS inhibits Thr495 phosphorylation and (2) phosphorylated CaMKII participated in Akt-dependent eNOS Ser1177 phosphorylation. Furthermore, NO is involved in the regulating activation of p38 MAPK-associated signaling cascade [36,37]. We investigated the effect of NO on p38 MAPK activation in eNOS^−/−^ mice to define the signaling pathway resulting from arginase II downregulation. As shown in Appendix A, treatment of arginase II inhibitor ABH activated CaMKII and p38 MAPK, and p38 MAPK inhibition did not affect CaMKII phosphorylation in the aortas of WT and eNOS^−/−^ mice. Therefore, p38 MAPK activation by arginase II downregulation did not depend on NO. Currently, we understood that arginase inhibition showed beneficial effect on NO production through increasing the substrate, L-arginine, bioavailability. Together with this suggestion, we, here, showed novel signaling cascade between arginase II inhibition and eNOS phosphorylation. This signaling pathway plays an important role in the preventative and therapeutic effects for vascular diseases arising from endothelial dysfunction. Additionally, this study indicates that mitochondrial dysfunction is associated with endothelial dysfunction because mitochondria as an organelle contributes to the functional regulation of cytosolic protein, eNOS in Ca^2+^-dependent manner.

The role of MAPK in eNOS phosphorylation was demonstrated in estrogen-treated and in polyphenols-treated ECs [35,38]. Interestingly, polyphenols treatment induced p38 MAPK/Akt-dependent phosphorylation of eNOS at Ser1177 and dephosphorylation of eNOS at Thr495 by a Ca^2+^-dependent phosphatase [38]. Consistent with this observation, we reported previously that polyphenols, such as piceatannol [39,40] and resveratrol [41], inhibited arginase II activity and that piceatannol potentiated eNOS Ser1177 phosphorylation [40]. Therefore, we suggest that eNOS activation is dependent on the Ca^2+^/CaMKII/AMPK/p38 MAPK/Akt signaling cascade. However, in disease-associated experiments, it has been shown that activated p38 MAPK negatively regulates eNOS activity [42,43]. Future studies on p38 MAPK isoforms, cellular localization, and upstream kinases are needed to address this discrepancy.

## 5. Conclusions

In conclusion, it has been shown that arginase II downregulation due to L-arg substrate bioavailability contributes to reciprocal regulation of eNOS activity [6,11,42,44]. It has also been shown that eNOS requires cofactors, such as BH_4_ and Zn^2+^, phosphorylation, and protein interactions for coupling. Here, we have proposed a novel signaling cascade for the arginase II-dependent increase in phosphorylation of eNOS at Ser1177 and decrease in phosphorylation of eNOS at Thr495. Arginase II downregulation increased [Ca^2+^]c in a p32-dependent manner that activated the CaMKII/AMPK/p38 MPAK/Akt/eNOS signaling pathway (Appendix A). In this signal transduction pathway, as a result of arginase II downregulation, p38 MAPK inhibition prevented enhanced NO production, decreased ROS generation, and augmented Ach-induced vasorelaxation. Therefore, p38 MAPK plays a crucial role in eNOS activation by arginase II downregulation.

## Figures and Tables

**Figure 1 cells-09-00392-f001:**
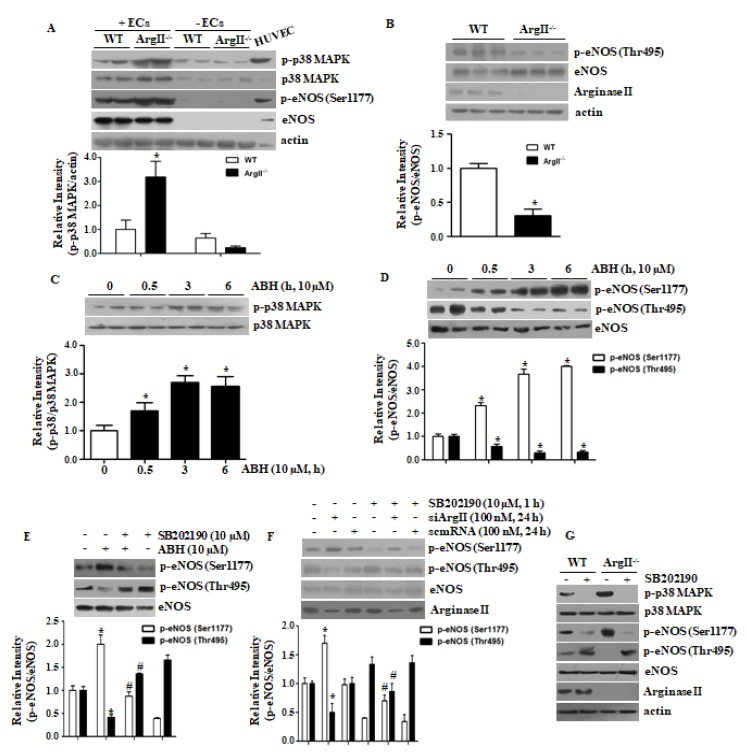
Arginase II downregulation induced p38 MAPK activation, which increased endothelial nitric oxide synthase (eNOS) phosphorylation at Ser1177 and decreased phosphorylation at Thr495 in HUVECs. Aortic vessels were isolated from wild-type (WT) and ArgII^−/−^ mice, and the lysates of intact (+ECs) and de-endothelialized (-ECs) aortas were analyzed. (**A**) Phosphorylation of p38 MAPK and eNOS Ser1177 was enhanced and; (**B**) phosphorylation of eNOS Thr495 was attenuated in the endothelium of aortas from ArgII^−/−^ mice. *n* = 4 and * vs. WT, *p* < 0.01; (**C**) Incubation with the arginase inhibitor 2(S)-amino-6-boronohexanoic acid (ABH) resulted in increased p38 MAPK phosphorylation and; (**D**) enhanced phosphorylation of eNOS at Ser1177 and decreased phosphorylation at Thr495 at the indicated time points. *n* = 4 and * vs. control (0 h), *p* < 0.05. Inhibition of p38 MAPK with SB202190 prevented phosphorylation of eNOS Ser1177 and enhanced phosphorylation at Thr495 in ABH-treated HUVECs (**E**) and in siArgII-treated HUVECs (**F**). * vs. untreated control, *p* < 0.01 and # vs. ABH or siArgII, *p* < 0.01, *n* = 3. The p38 MAPK inhibition with SB202190 blocked the phosphorylation of eNOS Ser1177 and augmented eNOS Thr495 phosphorylation in aortas of WT and ArgII^−/−^ (**G**).

**Figure 2 cells-09-00392-f002:**
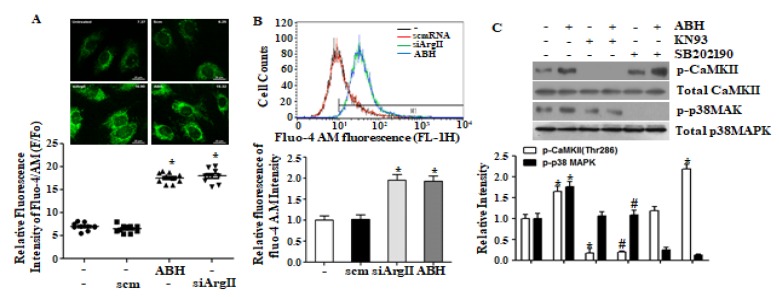
Arginase II downregulation-mediated increased [Ca^2+^]c elicited CaMKII-dependent p38 MAPK activation. (**A**) Microscopic and; (**B**) FACS analysis showed that arginase II downregulation with ABH or siArgII increased [Ca^2+^]c. Representative microscopic images are shown from 24 images from three experiments, and FACS analysis was performed four times. * vs. control, *p* < 0.01; (**C**) p38 MAPK activation by arginase II inhibition was blocked with the CaMKII inhibitor KN-93, but p38 MAPK inhibition with SB202190 had no effect on ABH-dependent CaMKII phosphorylation. * vs. untreated control, *p* < 0.01 and # vs. ABH, *p* < 0.05, *n* = 4 experiments.

**Figure 3 cells-09-00392-f003:**
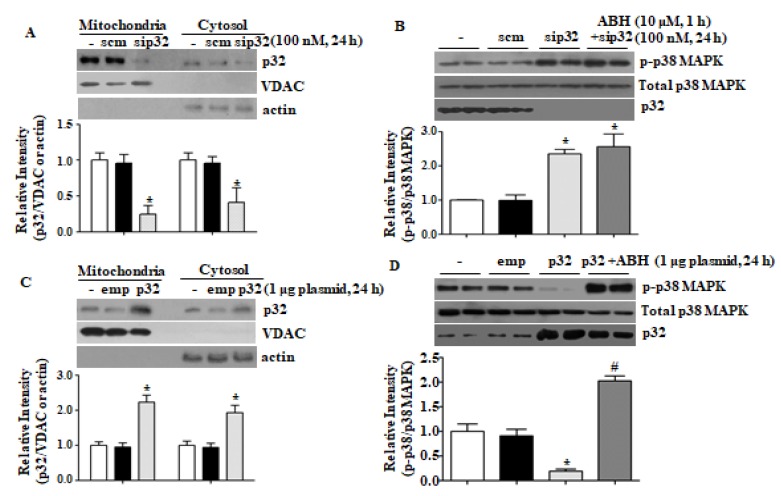
The p32 protein level regulated p38 MAPK activation. Treatment with sip32 reduced the p32 protein level (**A**) and activated p38 MAPK (**B**). * vs. control, *p* < 0.01, *n* = 3. Overexpression of p32 by plasmid transfection (**C**) reduced p38 MAPK phosphorylation that was restored with arginase inhibitor, ABH (**D**). * vs. control, *p* < 0.01, # vs. p32, *p* < 0.01, *n* = 4 experiments.

**Figure 4 cells-09-00392-f004:**
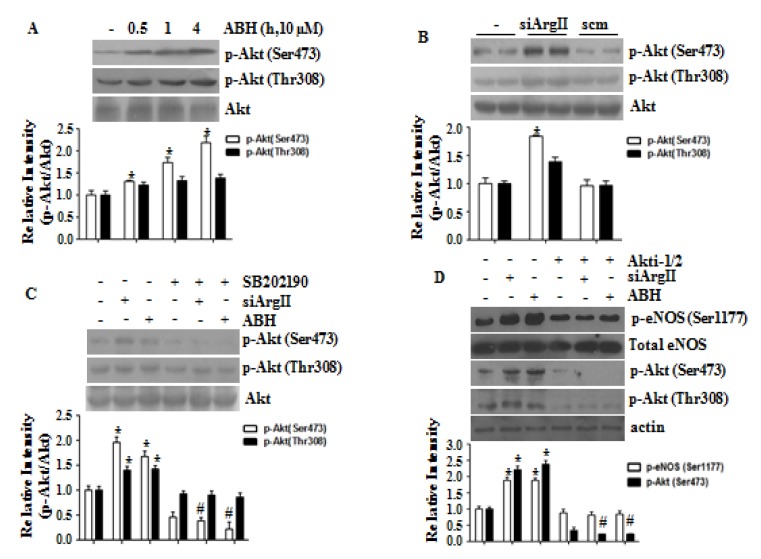
Arginase II downregulation induced p38 MAPK-dependent Akt activation that involved in phosphorylation of eNOS at Ser1177. The arginase inhibitor ABH (**A**) and siArgII (**B**) activated Akt, which was blocked by the p38 MAPK inhibitor SB202190 (**C**). (**D**) The Akt inhibitor Akti-1/2 prevented phosphorylation of eNOS at Ser1177. * vs. control, *p* < 0.01 and # vs. ABH or siArgII, *p* < 0.01, *n* = 3 experiments.

**Figure 5 cells-09-00392-f005:**
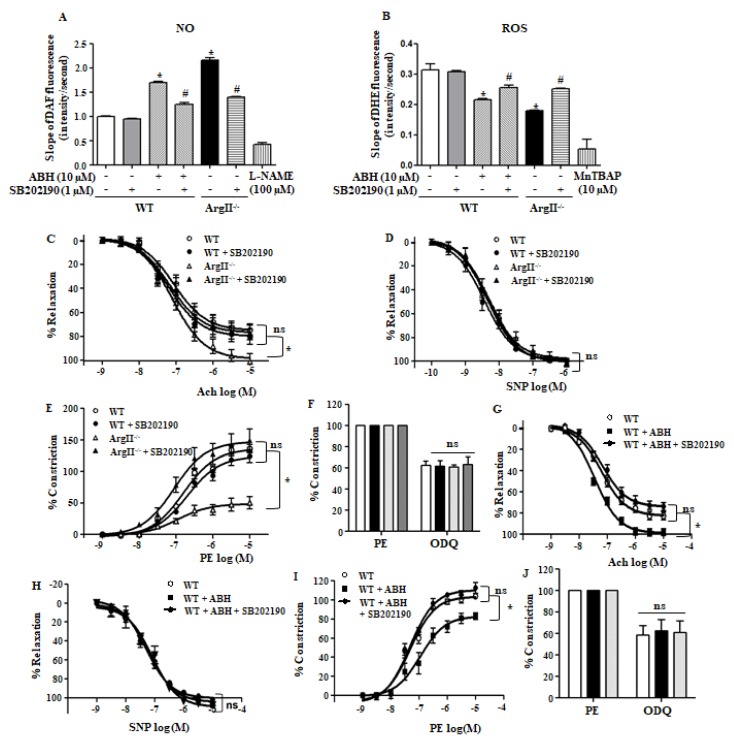
Arginase II inhibition enhanced nitric oxide (NO) production and decreased reactive oxygen species (ROS) generation in aortic endothelia in a p38 MAPK-dependent manner and p38 MAPK inhibition impaired endothelium-dependent vasorelaxation in aorta. (**A**) Increased NO production in aortic endothelia through arginase II inhibition or arginase II gene knockout (ArgII^−/−^) was prevented upon treatment with the p38 MAPK inhibitor SB202190; (**B**) Decreased ROS production caused by arginase II downregulation was augmented with the p38 MAPK inhibitor SB202190. * vs. control, *p* < 0.01 and # vs. ABH, *p* < 0.01, *n* = 4 experiments; (**C**) The enhanced Ach-induced vasorelaxation responses in aortic vessels of ArgII^−/−^ mice were attenuated with p38 MAPK inhibitor, but sodium nitroprusside (SNP) responses did not differ among the groups (**D**); (**E**) PE-induced constrictive responses were enhanced in p38 MAPK inhibitor-treated aortic vessels of ArgII^−/−^ mice; (**F**) sGC inhibitor 1H-[1,2,4]oxadiazolo[4,3-a]quinoxalin-1-one (ODQ) induced similar constrictive responses in all of the groups. *, *p* < 0.01, *n* = 8 aortic pieces from 3 mice; (**G**) Ach-dependent vessel relaxation caused by incubation of arginase inhibitor was blocked with p38 MAPK inhibitor without change in SNP responses (**H**); (**I**) PE-induced constrictive responses were augmented in SB202190-treated vessels; (**J**) ODQ responses did not differ. * *p* < 0.01, *n* = 8 aortic pieces from 3 mice.

**Figure 6 cells-09-00392-f006:**
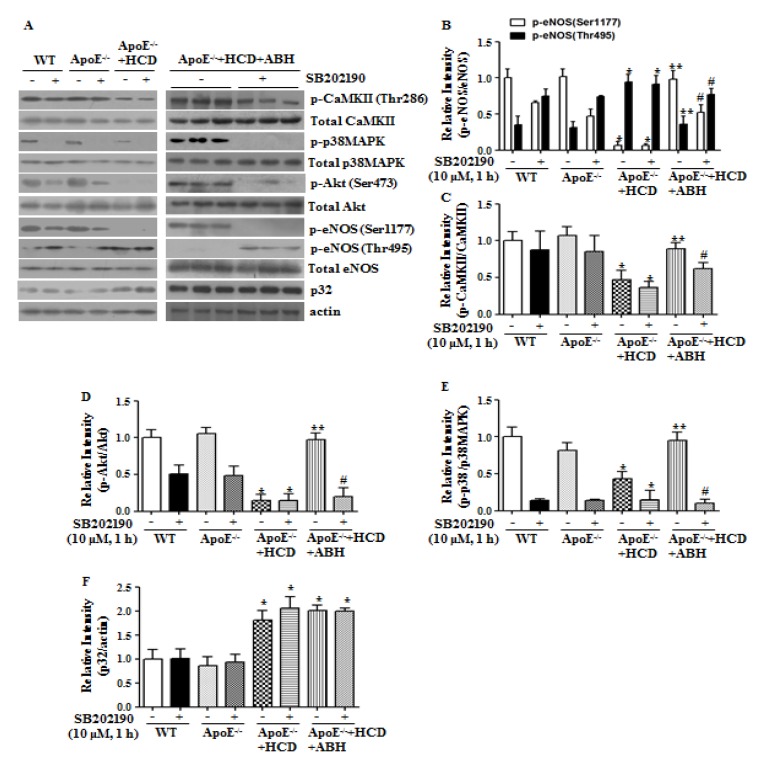
The p38 MAPK inhibition attenuated the enhanced eNOS phosphorylation at Ser1177 caused by arginase inhibition in ApoE^−/−^ mice fed a high cholesterol diet (HCD). In aortas of ApoE^−/−^ mice fed an HCD, the impaired CaMKII/p38 MAPK/Akt/eNOS signaling cascade was restored upon arginase inhibition. However, p38 MAPK inhibitor blocked the restoration of the signal transduction pathway (**A**–**F**). The bar graphs show the results of densitometric analyses from 3 independent experiments. * vs. WT, *p* < 0.01; ** vs. ApoE^−/−^ + HCD, *p* < 0.01; and # vs. ApoE^−/−^ + HCD + ABH, *p* < 0.01.

**Figure 7 cells-09-00392-f007:**
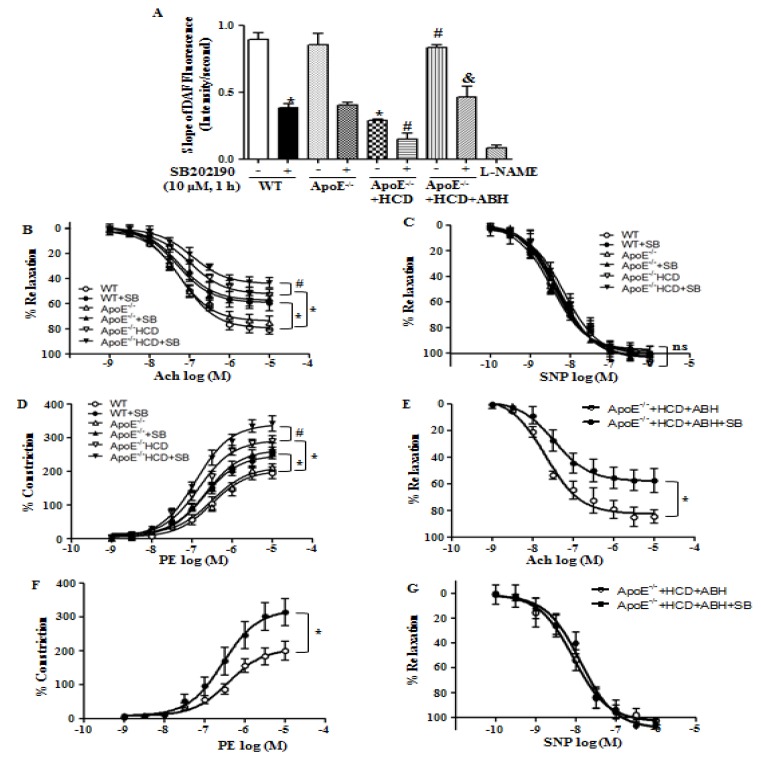
The p38 MAPK inhibition accelerated endothelial dysfunction in ApoE^−/−^ mice fed an HCD. (**A**) Arginase II inhibition restored the NO production that was impaired by p38 MAPK inhibitor in ApoE^−/−^ mice fed an HCD. * vs. WT, *p* < 0.01; ** vs. ApoE^−/−^ + HCD, *p* < 0.01; # vs. ApoE^−/−^ + HCD + ABH, *p* < 0.01; (**B**) Arginase inhibitor ABH enhanced Ach-dependent relaxation responses in aortic vessels of ApoE^−/−^ mice fed an HCD without a difference in SNP-dependent vasorelaxation (**C**); (**D**) PE-induced constrictive responses were augmented with treatment of p38 MAPK inhibitor, SB202190. The restored responses to Ach (**E**) and PE (**F**) through treatment of arginase inhibitor were impaired with incubation of p38 MAPK inhibitor, SB202190 without a difference in SNP responses (**G**). * *p* < 0.01 and # *p* < 0.01, *n* = 9 aortic pieces from 4 mice.

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
