# Peer review of "p32-Dependent p38 MAPK Activation by Arginase II Downregulation Contributes to Endothelial Nitric Oxide Synthase Activation in HUVECs"

_cells, 2020, doi:10.3390/cells9020392_

Round 1
Reviewer 1 Report
This is an interesting report on the novel mechanism of regulation of eNOS activity upon downregulation of arginase II. The Comprehensive work is well documented and conclusions are supported by experimental results. Clarity of results description could be improved. certain parts of the manuscript are rather cumbersome and difficult to follow. On Fig. 1a the effect of de-endothelialization on the level of aortas p-p38MAPK and p-eNOS(Ser1177) are shown, however the WB of nascent e-NOS or p38MAPK can not be found.
Author Response
This is an interesting report on the novel mechanism of regulation of eNOS activity upon downregulation of arginase II. The Comprehensive work is well documented and conclusions are supported by experimental results. Clarity of results description could be improved. certain parts of the manuscript are rather cumbersome and difficult to follow. On Fig. 1a the effect of de-endothelialization on the level of aortas p-p38MAPK and p-eNOS(Ser1177) are shown, however the WB of nascent e-NOS or p38MAPK can not be found.
; Thanks for the comment. We added the control blots, total eNOS and p38 MAPK to Fig 1A
Reviewer 2 Report
The authors here introduced a novel mechanistic studies on how A2 deletion activate eNOS using genetic and pharmacological models
1-Material and methods sections, what is the source of mice, were the WT and A2-/- bred and housed -under the same conditions
2-Results.
2a-Fig 1A, can the authors normalize p-p38 expression to total p38 and penos to total enos
2b-What is the effect of wt and a2-/- on p-p38 and p-enos in the presence of p38 inhibitor
2c-Fig.3. what is the effect of sip32 and p32 overexpression on penos
3- English and grammer revisions are required
-Introduction line 60, Arginase dependent NO production….It is better to say decline in NO production.
-Line68, With together…better to say additionally
-Line 327, please revise …………………… we reported that p32 played a important protein
-Line 352 please revise………………. domain to oxygenase domain by replacing by displacing the
4-Can the authors add the translational impact of this study
5-A recent study showed that A2-/- protected against high fat diet-induced systemic oxidative stress and endothelial dysfunction (Atawia, R.T., IJMS,2019) can supportyour findings and discussion section
Reviewer 3 Report
Broad comments
Introduction part must describe the rational to select Akt signaling & eNOS-Akt phosphorylation sites selected in this study. Line 49-50- structure of this sentence appears incomplete without referring the mentioned study. Figure 1E – in the absence of any cellular stress the expression on p-p38MAPK looks high in HUVEC cells, are these cells stimulated with any drug? Or p-p38MAPK constitutively activated in HUVEC cells, independent of stimulation? Figure 1E- how did authors confirm 1uM SB202190 is sufficient to inhibit p-eNOS 1177 and what is the significance difference between untreated and only SB202190 treated cells for p-eNOS1177 expression? Figure 1E & F- Rational to select two different concentration of SB202190 is not clear. Figure 2A- nothing can be interpreted from the following microscopic resolution, the bottom two pictures looks highly saturated. Figure 2C- SB202190 is a catalytic inhibitor of p38MAPK, why the expression of p38MAPK reduced (lane 5) in SB202190 treated cells? Figure 3B- need to provide evidence that silencing p32 in HUVECs has direct effect on Arginase II signaling, in Figure 3D-whether ABH restored p-p38MAPK expression in sip32 treated cells?? Figure 3- what is the status of cyto-mito calcium ions concentration in this set-up of experiment? Figure 5- explain Ach, SNP, sGC, PE in this study. As most of the studies conducted with siRNA, it is important to validate some of the important data with a different set of siRNA. Also include RT-qPCR data for knockdown efficacy of Arg and p32.Specific comments
Catalogue number of antibodies used in this study for the better reproducibility of results. Figure 1D, 3A & 3C- bands were cropped too closed, please provide some space at the end.Author Response
Please check attached file

Round 2
Reviewer 2 Report
Grammar and English language revisions are required
for example
line 70, ....lead to
line 71,.... full activation and augmentation
line 101..... were bred
Reviewer 3 Report
This revised version of the manuscript is very much improved with some additional data sets, I have no further comment.